# LEARNING AND MEMORIZATION (EXTENDED ABSTRACT)

**Satrajit Chatterjee**
Two Sigma
{satrajit.chatterjee}@twosigma.com

## ABSTRACT

In the machine learning research community, it is generally believed that there is a tension between memorization and generalization. In this work we examine to what extent this tension exists, by exploring if it is possible to generalize through memorization alone. Although direct memorization with a lookup table obviously does not generalize, we find that introducing depth in the form of a network of support-limited lookup tables leads to generalization that is significantly above chance and closer to those obtained by standard learning algorithms on several tasks derived from MNIST and CIFAR-10. Furthermore, we demonstrate through a series of empirical results that our approach allows for a smooth tradeoff between memorization and generalization and exhibits some of the most salient characteristics of neural networks: depth improves performance; random data can be memorized and yet there is generalization on real data; and memorizing random data is harder in a certain sense than memorizing real data. The extreme simplicity of the algorithm and potential connections with stability provide important insights into the impact of depth on learning algorithms, and point to several interesting directions for future research.

## 1 INTRODUCTION

Neural networks trained through stochastic gradient descent (SGD) are capable of memorizing their training data. Although practitioners have long been aware of this phenomenon, Zhang et al. (2016) recently brought attention to it by showing that standard SGD-based training on AlexNet gets close to zero training error on a modification of the Imagenet dataset even when the labels are randomly permuted. This leads to an interesting question: *If nets have sufficient capacity to memorize random training sets why do they generalize on real data?* A natural hypothesis is that nets behave differently on real data than on random data. Arpit et al. (2017) study this question experimentally and show that there are apparent differences in behavior. They conclude that generalization and memorization depend not just on the network architecture and optimization procedure but on the dataset itself.[1]

But what if networks fundamentally do *not* behave differently on real data than on random data, and, in both cases, are simply memorizing? This is a difficult question to entertain for two reasons. First, it is hard to provide a direct answer. Whereas it is easy to tell when a net is memorizing random data (the training error goes to zero!), there is no easy way to tell when a network is memorizing real data (as opposed to learning).[2] Second, and perhaps more importantly, it contradicts the intuitive notion —inherent in the preceeding discussion—that memorization and generalization are at odds.

This work attempts to shed light on this second difficulty by investigating the following: *How much can you learn if memorization is all you can do? Is generalization even possible in this setting?*

---

[1]One might be tempted to think that the result of Zhang et al. and the question above is in contradiction with the results of conventional statistical learning theory but that is not the case: see Kawaguchi et al. (2017) for a detailed discussion.

[2]Indeed, Arpit et al. (2017) view one of their main contributions to be the argument that the operational definition of memorization is the behavior on random data (see the Introduction of their paper).

Table 1: The test accuracy of memorization is significantly better than chance on Binary-MNIST (left) and on Binary-CIFAR-10 (right) and is closer to those of several standard methods. Memorization ties with nearest neighbors on CIFAR. A LeNet-style CONV NET does the best in both cases but unlike the others it is not permutation invariant.

| METHOD | ACCURACY | | METHOD | ACCURACY | |
|---|---|---|---|---|---|
| | TRAINING | TEST | | TRAINING | TEST |
| CONV NET | 0.98 | 0.98 | CONV NET | 0.93 | 0.71 |
| 5-NEAREST NEIGHBORS | 0.99 | 0.97 | RANDOM FOREST (300 TREES) | 1.00 | 0.66 |
| 1-NEAREST NEIGHBOR | 1.00 | 0.97 | 5-NEAREST NEIGHBORS | 0.75 | 0.63 |
| RANDOM FOREST (10 TREES) | 1.00 | 0.96 | 1-NEAREST NEIGHBOR | 1.00 | 0.63 |
| **MEMORIZATION** ($k = 12$) | **0.99** | **0.90** | **MEMORIZATION** ($k = 10$) | **0.79** | **0.63** |
| LOGISTIC REGRESSION | 0.87 | 0.87 | LOGISTIC REGRESSION | 0.64 | 0.56 |
| NAIVE BAYES | 0.76 | 0.77 | NAIVE BAYES | 0.55 | 0.56 |
| RANDOM GUESS | 0.50 | 0.50 | RANDOM GUESS | 0.50 | 0.50 |

## 2 THE MODEL

To make our investigation concrete, we shall focus on learning binary classifiers on Boolean inputs.[3] For example, consider the Binary-MNIST task of classifying 1-bit quantized MNIST images into two classes (0-4 v/s 5-9). The simplest way to memorize would be to build a lookup table ("lut") from the training data, but that obviously does not generalize. We could do better with $k$-Nearest Neighbors, but on many problems it is not easy to construct semantically meaningful distance functions on the input domain. Indeed, a key result in deep learning is the notion of a learned embedding.

Instead, we appeal to the notion of depth which has been wildly successful in improving the performance of neural networks. Instead of building a single large lut, we build a network of (much) smaller luts that are arranged in layers like neurons in a deep neural network. Because a lut, unlike a neuron, can implement an arbitrary function, for depth to be useful, we need to limit the complexity. We do that simply by limiting the support of the lut. Thus each lut receives inputs from $k$ luts in the previous layer (or $k$ inputs for luts in the first layer) which are picked at random when the network is constructed. In our experiments $k$ is typically 10 or less.

Training is done through memorization and proceeds layer by layer from inputs to outputs. Each lut in a layer memorizes the mapping from its input bit patterns to the final target, i.e., it constructs a function $\hat{f} : \{0, 1\}^k \to \{0, 1\}$ where each possible bit pattern $p \in \{0, 1\}^k$ at the input of the lut gets mapped to the class $y \in \{0, 1\}$ that is most commonly associated with it in the training set.[4] Ties (which include as a special case patterns not seen in training) are broken randomly when constructing $\hat{f}$. Each lut then uses its learned $\hat{f}$ to map its inputs to outputs. The outputs of a layer form the inputs for the next layer which in turn learns the mapping from this new representation to the final target. Inference proceeds in a similar fashion from inputs to outputs.

---

[3]This is less restrictive than it appears given the progress towards quantized networks (e.g. Rastegari et al. (2016)) and even in real valued networks we need convolution and pooling for certain kinds of continuity.

[4]$\hat{f}$ is stored as a truth table.

Table 2: Layer by layer training accuracy of network of 8-input lookup tables on Binary-MNIST. Layer 0 is the input. Test accuracy is 0.87.

| LAYER | COUNT | TRAINING ACCURACY | | | |
|---|---|---|---|---|---|
| | | MEAN | STD | MIN | MAX |
| 0 | 784 | 0.51 | 0.0340 | 0.40 | 0.66 |
| 1 | 1024 | 0.61 | 0.0403 | 0.51 | 0.73 |
| 2 | 1024 | 0.74 | 0.0191 | 0.67 | 0.79 |
| 3 | 1024 | 0.83 | 0.0068 | 0.80 | 0.85 |
| 4 | 1024 | 0.87 | 0.0033 | 0.86 | 0.88 |
| 5 | 1024 | 0.88 | 0.0015 | 0.88 | 0.88 |
| 6 | 1 | 0.89 | 0.0000 | 0.89 | 0.89 |

Table 3: Effect of varying lookup table size on Binary-MNIST.

| $k$ | ACCURACY | | | |
|---|---|---|---|---|
| | ON REAL DATA | | ON RANDOM DATA | |
| | TRAINING | TEST | TRAINING | TEST |
| 2 | 0.66 | 0.66 | 0.51 | 0.53 |
| 4 | 0.81 | 0.81 | 0.53 | 0.54 |
| 6 | 0.85 | 0.86 | 0.55 | 0.52 |
| 8 | 0.89 | 0.87 | 0.60 | 0.51 |
| 10 | 0.94 | 0.89 | 0.69 | 0.50 |
| 12 | 0.99 | 0.90 | 0.82 | 0.51 |
| 14 | 1.00 | 0.83 | 0.92 | 0.51 |
| 16 | 1.00 | 0.66 | 0.98 | 0.52 |

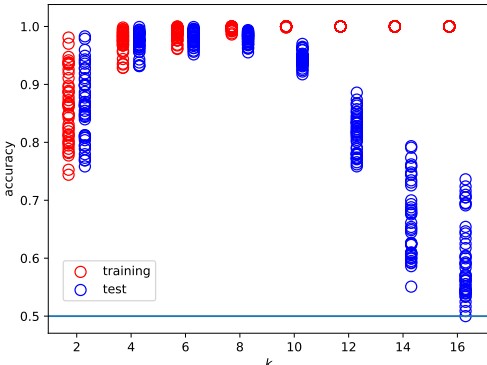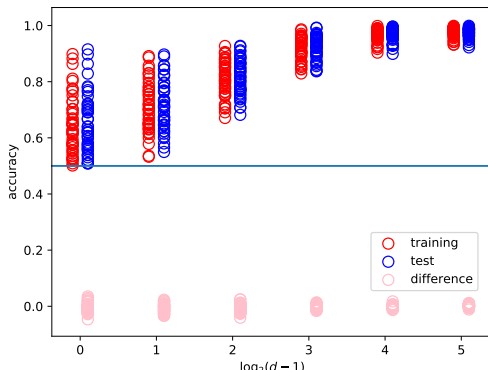

Figure 1: Generalization error (difference of training and test accuracy) goes up as $k$ increases on the 45 pairwise separation tasks on 1-bit quantized MNIST (left). The large variation for $k = 2$ is due to insufficient mixing, and as we increase the depth of the network training this goes down (right).

If we measure training error as 0-1 loss, the function $\hat{f}$ learned by a lut is (Bayes) *optimal*, i.e., there is no other function that has strictly lower training error. It is also *monotonic*: if we make the lut wider by providing it with more inputs, this procedure cannot increase training error. It is important to note that these properties hold "locally" for a lut but not for the network as a whole.

This is pure memorization in the sense that unlike most learning algorithms (with the important exception of $k$-NNs discussed previously) we do not solve any optimization problem to figure out the trainable parameters of the network. Furthermore, the algorithm is extremely computationally efficient and easy to parallelize since it relies only on counting and dense table lookups.

## 3 RESULTS

Our main result is that generalization is possible with pure memorization in the sense that memorization achieves test accuracies significantly above chance (0.5). We see this on Binary-MNIST and Binary-CIFAR-10 (Table 1) and in the 45 pairwise separation tasks of MNIST (Figure 1) and of CIFAR-10 (Table 4). In all these cases we use a baseline network with 5 hidden layers of 1024 luts (and an output layer with a single lut) with $k$ as indicated. Furthermore, these results are very stable w.r.t. choices made at network construction. Table 2 along with optimality and monotonicity properties (Section 2) provides some intuition for that.

The size of each lut ($k$) provides a way to tradeoff memorization and generalization (Table 3 and Figure 1). Also, like the experiments of Zhang et al. (2016) we are able to memorize random data and yet generalize on real data; and like Arpit et al. (2017) we find a sense in which memorizing random data is harder (Table 3).

Finally, it is interesting to speculate why support limited memorization generalizes. The better generalization of smaller luts provides a hint: The same number of training examples are divided over fewer rows which makes the luts more likely to be stable in the sense of Bousquet & Elisseeff (2002). By thinking of the problem in these terms it seems natural that some forms of memorization may generalize well, and this could lead the way to more computationally efficient learning algorithms that memorize in a stable manner.

Table 4: Training (below diagonal) and test accuracy (above) on Pairwise-CIFAR-10 ($k = 10$).

|  | PLANE | AUTO | BIRD | CAT | DEER | DOG | FROG | HORSE | SHIP | TRUCK |
|---|---|---|---|---|---|---|---|---|---|---|
| PLANE |  | 0.77 | 0.77 | 0.80 | 0.82 | 0.81 | 0.84 | 0.81 | 0.71 | 0.79 |
| AUTO | 0.96 |  | 0.79 | 0.77 | 0.79 | 0.80 | 0.80 | 0.78 | 0.76 | 0.67 |
| BIRD | 0.95 | 0.98 |  | 0.68 | 0.63 | 0.69 | 0.66 | 0.72 | 0.83 | 0.81 |
| CAT | 0.96 | 0.98 | 0.96 |  | 0.70 | 0.61 | 0.68 | 0.71 | 0.81 | 0.76 |
| DEER | 0.96 | 0.98 | 0.95 | 0.96 |  | 0.73 | 0.69 | 0.71 | 0.83 | 0.81 |
| DOG | 0.98 | 0.99 | 0.97 | 0.96 | 0.97 |  | 0.72 | 0.70 | 0.82 | 0.79 |
| FROG | 0.97 | 0.98 | 0.95 | 0.95 | 0.97 | 0.96 |  | 0.75 | 0.85 | 0.80 |
| HORSE | 0.98 | 0.99 | 0.96 | 0.97 | 0.96 | 0.98 | 0.97 |  | 0.81 | 0.75 |
| SHIP | 0.93 | 0.96 | 0.98 | 0.98 | 0.98 | 0.98 | 0.99 | 0.98 |  | 0.77 |
| TRUCK | 0.97 | 0.95 | 0.98 | 0.97 | 0.97 | 0.98 | 0.98 | 0.97 | 0.98 |  |

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
