# OpenReview forum: "Learning and Memorization"
_ICLR.cc/2018/Workshop — Accept_

### Official Review · AnonReviewer3 · 2018-03-09
**Exploring the generalization and memorization in deep nets but lack contribution and clarity**

**Rating:** 4
**Confidence:** 4

**Review:**

This paper investigates the memorization and generalization problems and examines the possibility of generalization through memorization alone.
This work touches very interesting and important problem. However, the model/experiments proposed in the paper is very ambiguous and unclear to me. In addition, it is not clear to me what is the contribution here.
Another issue is that the main claim of the paper, generalization is possible with pure memorization, in the paper is not conclusive due to that fact that the experiments are done on Binary-MNIST and Binary-CIFAR-10 and in my view, these experiments are not thorough enough to reach this conclusion.

Given the focus of the workshop track which is late-breaking development, very novel ideas, or position paper,  I am not convinced this paper is a good fit for the workshop track.

---

### Official Review · AnonReviewer1 · 2018-03-09
**Great experiment**

**Rating:** 9
**Confidence:** 4

**Review:**

In this paper the author proposes a simple experiment: they build a network of support-limited lookup tables, structured like deep dense neural networks, and show that such a simple memorization mechanism is still able to generalize, in the sense that it performs much better than chance on a test set.

Not only is this paper well-written, it performs a simple experiment that should, once again, make us reconsider many of our preconceived notions regarding neural networks.
Since it is fairly self-contained, I don't have many remarks:
- What is the distribution of the random data that you use? (I imagine bits with p=1/2)
- Does randomly breaking ties/missing data make a difference compared to outputting a constant? (e.g. 0)
- You compare to a convnet for completeness, but I don't think that's fair given the geometrical prior. How hard would it be to implement a "lookup-table-convnet"?

---

### Official Review · AnonReviewer2 · 2018-03-11
**Interesting model, merits discussion**

**Rating:** 8
**Confidence:** 5

**Review:**

The paper proposes an interesting way of building a hierarchy of features using memorization alone. Each feature is a lookup table that maps k-bit binary strings to {0, 1} corresponding to the two classes in a binary classification problem. The entry for a particular k-bit string is 0 or 1 depending on the majority class among data points that lead to that k-bit string as the input. The model consists of layers of such features where each feature looks at a random subset of k features from the preceding layer.

This model is reminiscent of a cascade of random forests. It has the interesting property of being deep and non-linear and at the same time very easy to construct by memorization in a layer-by-layer manner. While the results are not great, given the simplicity of the model, they are promising enough to merit discussion and more investigation.

---

### Decision · Program_Chairs · 2018-03-20
**ICLR 2018 Workshop Acceptance Decision**

**Decision:**

Accept

**Comment:**

Congratulations, your paper was accepted to the ICLR workshop.